# Encapsulation of Marjoram Phenolic Compounds Using Chitosan to Improve Its Colon Delivery

**DOI:** 10.3390/foods11223657

**Published:** 2022-11-16

**Authors:** María de las Nieves Siles-Sánchez, Laura Jaime, Marisol Villalva, Susana Santoyo

**Affiliations:** Institute of Food Science and Research (CIAL), Universidad Autónoma de Madrid (CEI UAM + CSIC), 28049 Madrid, Spain

**Keywords:** *Origanum majorana*, phenolic compounds, nano/microparticles, colon delivery systems, ionic gelation, spray drying

## Abstract

In this study, chitosan particles were used to encapsulate marjoram phenolic compounds as colon-specific drug-delivery systems. The microparticles were prepared by ionic gelation and spray-drying techniques and varying amounts of polymer and extract, along with different method conditions. The spray drying of microparticles (0.75% low molecular weight chitosan dissolved in 0.4% of acetic acid) presented the best encapsulation efficiency (near 75%), with size ranges from 1.55 to 1.68 µm that allowed the encapsulation of 1.25–1.88 mg/mL of extract. Release studies of individual marjoram phenolic compounds at pH 2 and 7.4 showed that most of the compounds remained encapsulated in the microparticles. Only arbutin and vicenin II presented a high initial burst release. As the polarity of the compounds was reduced, their initial release decreased. In addition, after gastrointestinal digestion, most of marjoram phenolic compounds remained encapsulated. These results prove that chitosan microparticlescould protect the marjoram phenolic compounds during gastrointestinal digestion, specifically those related to anticancer activity, which enables their application as colon-specific delivery systems.

## 1. Introduction

Colon-specific drug-delivery systems via oral administration have become an important area of research in last few years, due to the great advantages this route provides in the treatment of several colon diseases (i.e., ulcerative colitis, Crohn’s disease, and colorectal cancer). However, these systems must prevent the release of the incorporated therapeutic agents in the upper gastro-intestinal tract following their oral administration, while allowing their release when they arrive at the colon [1]. 

In this context, several natural and synthetic polymers have been studied in order to develop colon-specific drug-carrier vehicles. Among these polymers, chitosan and its derivates have gained enormous attention for usage in various fields [2], including colonic drug delivery [3]. Chitosan is a cationic polysaccharide produced by the deacetylation of chitin that consists of a linear co-polymer of β-(1→4)-N-acetyl-D-glucosamine with β-(1→4)-D-glucosamine ramifications. Chitosan can maintain its integrity in the upper gastrointestinal tract, being degraded in the colon by the action of enzymes produced by colonic microorganisms. In addition, chitosan presents mucoadhesive properties, as its positively charged amino groups attach to negative charges from sialic acid present in the mucus surface via electrostatic interaction, increasing its residence time within colon epithelia [4].

Micro- and nanoparticles formulated using chitosan have been reported as colon-specific drug vehicles. Samprasit et al. [5] showed the potential of alpha-mangostin and resveratrol containing chitosan nanoparticles prepared by ionotropic gelation as an orally deliverable colon-cancer formulation. Sabra et al. [6] also proposed chitosan-pectinate nanoparticles for the colon-targeted delivery of curcumin, prepared by the ionic gelation method. In addition, dexamethasone-loaded chitosan microparticles prepared by spray drying presented a potential use as a means of colon-specific drug delivery [7].

*Origanum majorana* L., commonly known as marjoram, is a herb that is widely used in traditional medicine in the Mediterranean region. Different studies have shown that it exerts biological activities as an antimicrobial, anti-inflammatory, antioxidant, and antiproliferative agent [8]. In this regard, marjoram phenolic compounds have been frequently related to these activities. Mainly phenolic acids, such as rosmarinic acid, as well as other phenolic compounds such as luteolin, apigenin, and apigenin’s derivatives [9,10,11] have also been indicated as potential agents against colorectal cancer. Accordingly, Venkatachalam et al. [12] reported that rosmarinic acid could act as a potent chemopreventive agent in colon cancer, as it prevented the formation and multiplicity of aberrant crypt foci, protected cells from the damage caused by 1,2-dimethylhydrazine and its metabolites, spared the activities of antioxidant enzymes, and induced the apoptosis of tumor cells. In addition, Redondo-Blanco et al. [13] indicated that apigenin induced growth inhibition, cell cycle arrest, and apoptosis in colon cancer cell lines. Luteolin has been also proposed as a therapeutic agent in colorectal tumors [14]. 

The aim of the present work was to prepare chitosan micro/nanoparticles to increase the colon-specific delivery of marjoram phenolic compounds. Micro/nanoparticles were prepared using low- and medium-molecular-weight chitosan and two different technological processes: ionic gelation and spray drying. Particles were characterized in terms of yield, size, a polydispersity index, and encapsulation efficiency. In addition, in vitro release studies of individual marjoram phenolic compounds at pH 2 and 7.4 were performed to elucidate the ability of the developed formulations to retain the different phenolic compounds at gastrointestinal pHs. Finally, an in vitro gastrointestinal digestion was also carried out to study the combined effect of pH and gastrointestinal enzymes on the developed formulations.

## 2. Materials and Methods

### 2.1. Materials

Low-molecular-weight chitosan (LCH, viscosity 108 cP, 1% in 1% acetic acid, deacetylation degree 83%), medium-molecular-weight chitosan (MCH, viscosity 407 cP, 1% in 1% acetic acid, deacetylation degree 75%), and pentasodium tripolyphosphate (TPP) were purchased from Sigma-Aldrich (Madrid, Spain).

Ethanol (99.5% purity) was purchased from Panreac (Barcelona, Spain). Formic acid (99%) was obtained from Acros Organics (Madrid, Spain) and acetonitrile HPLC grade was obtained from Macron Fine Chemicals (Madrid, Spain). Phenolic compounds standards (HPLC purity ≥95%), such as rosmarinic acid and eriodyctiol, were obtained from Sigma-Aldrich (Madrid, Spain). Lithospermic acid, salvianolic acid, apigenin, apigenin 7-O-β-glucuronide, and vicenin II were obtained from Phytolab (Madrid, Spain). Ethyl gallate, apigenin 7-O-glucoside, caffeic acid, and luteolin 7-O-β-glucuronide were obtained from Extrasynthese S.A. (Genay, France); arbutin was obtained from TCI (Belgium); and taxifolin was obtained from Fluka analitycal (Madrid, Spain).

### 2.2. Methods

#### 2.2.1. Plant Material and Ultrasound Assisted Extraction (UAE) 

Dried leaves of *Origanum majorana* L. were obtained from a Spanish company that specializes in plant material supplies (Murciana Herboristería, Murcia, Spain). The leaves were ground in a knife mill (Grindomix GM 200, Retsch, Llanera, Spain) and the particle size was reduced to <500 µm. Ultrasound-assisted extraction (UAE) was carried out, following the method described in Villalva et al. [15], with slight modifications. Briefly, a Branson 250 digital device (Branson Ultrasonic, Danbury, USA) with an electric power of 200 W and a frequency of 60 Hz was used. Twenty g of ground plant material was extracted with ethanol, maintaining a ratio of 1:10 (plant:solvent) for 20 min and 60% amplitude using a probe one-half inch in diameter. Next, the samples were filtered and ethanol was removed under vacuum at 35 °C (IKA RV 10, Madrid, Spain).

#### 2.2.2. Preparation of Chitosan Micro/Nanoparticles Containing Marjoram Extract

##### Ionic Gelation

Chitosan particles were obtained by inducing the gelation of a chitosan solution with TPP, following the method described by Da Silva et al. [16], with adaptations. Once the LCH or the MCH (1% *w*/*v*) were dissolved in the acetic acid solution (1% *v*/*v*), marjoram extract was added at different final concentrations (3, 2, 1, 0.5, and 0.25 mg/mL) and the pH value was adjusted to 5.8 with 1 M sodium hydroxide. Then, TPP was dissolved in milli-Q water at 0.1% (*w*/*v*) and, in order to study the best ratio for chitosan:TPP, different volumes of the TPP solution were added to the chitosan solution under magnetic stirring at room temperature for 20 min. The ratios of chitosan:TPP assayed were 5:1, 6:1, and 7:1. After that, the suspension was centrifugated (40,000× *g*, 45 min, T = 25 °C) and the supernatant was discarded to obtain the particles. The particles were frozen, then freeze-dried (LyoBeta 15, Telstar, Spain) and stored at 4 °C until analyses were performed.

##### Spray Drying

Microparticles of LCH or MCH containing marjoram extract were prepared by spray drying according to the method of Cabral et al. [17], with adaptations. First, the optimal ratio of chitosan:acetic acid was studied. Then, different chitosan concentrations (2, 1, 0.75, and 0.5% *w*/*v*) and acetic acid (1, 0.5 and 0.4% *v*/*v*) were used to determine the best ratio. After that, different marjoram ethanolic extract concentrations were added and mixed for 2 h. The mass ratios of chitosan:extract that were studied were 4:1, 6:1, and 8:1 (*w*/*w*).

All solutions were submitted to spray dryer equipment (Mini büchi B-191, Switzerland) at a feed rate of 4 mL/min. Inlet and outlet air temperatures and aspiration rates were set to 170 °C, ≈70 °C, and 60%, respectively. The microparticles obtained were stored at 4 °C until the experiments were carried out.

#### 2.2.3. Analysis of Phenolic Composition and Quantification by HPLC-PAD 

Chromatographic analyses were carried out as described previously by Villalva et al. [10]. Briefly, an Agilent HPLC 1260 Infinity series system with a PAD system (Agilent Technologies Inc., Santa Clara, CA, USA) equipped with an ACE Excell 3 Super C18 column (150 mm × 4.6 mm, 3 μm particle size), protected by a guard column ACE 3 C18-AR (10 mm × 3 mm) at 35 °C, was used. 

The mobile phase consisted of solvent A (99.9/0.1 water/formic acid *v*/*v*) and solvent B (acetonitrile) at a constant flow rate of 0.5 mL/ min, following the next gradient elution: 0 min, 0% B; 1 min, 0% B; 6 min, 15% B; 21 min, 25% B; 26 min, 35% B; 36 min, 50%B; 41 min, 50% B; 43 min, 100% B; 53 min, 100% B; 55 min, 0% B; and 60 min, 0% B. The sample was centrifuged with an Amicon filter 3kDa (VWR, USA) at 16,000× *g* for 45 min and filtered by a 0.45 μm PVDF filter prior injection (20 µL). The detection wavelengths were 280, 320, and 360 nm. Quantification of the identified compounds was carried out using calibration curves of pure analytical standards.

#### 2.2.4. Characterization of the Chitosan Micro/Nanoparticles Containing Marjoram Extract

##### Determination of Particle Size Distribution, Zeta Potential, and Polydispersity Index (PDI)

The average particle size, zeta potential, and polydispersity index were determined by a Zeta-sizer Ultra (Malvern Instruments Ltd., Malvern, UK). For analysis, the particles were dispersed in ultrapure grade water at a 1:500 ratio. All experiments were performed in triplicate.

##### Determination of Yield and Encapsulation Efficiency (EE)

The yield of the process for both formulations was calculated as follows (Equation (1)):(1)$$Yield\;\left(\%\right)=\left(\frac{weight\;of\;the\;micro/nanoparticles\;obtained}{weight\;of\;solids\;added\;to\;prepare\;the\;micro/nanoparticles}\right)\times 100$$

The encapsulation efficiency of the marjoram phenolic compounds in micro/nanoparticles obtained by ionic gelation was determined in the supernatant after particle centrifugation (40,000× *g*, 45 min). Supernatants were recovered and recentrifuged (16,000× *g*, 45 min) using an Amicon filter 3kDa (VWR, USA), with prior analysis by HPLC. Thereupon, the non-encapsulated compounds were determined and the EE % was calculated using Equation (2). The encapsulation efficiency of individual phenolic compounds (individual EE%) was also calculated in the supernatant using Equation (3).
(2)$$Total\;EE\;\left(\%\right)=100-\left(\frac{\Sigma \;of\;the\;supernatant\;phenolic\;compounds}{\Sigma \;of\;the\;phenolic\;compounds\;in\;extract}\right)\times 100$$
(3)$$Individual\;EE\;\left(\%\right)=100-\left(\frac{\mathrm{Individual}\;\mathrm{phenolic}\;\mathrm{compound}\;\mathrm{in}\;\mathrm{supernatant}}{Individual\;phenolic\;compound\;quantified\;in\;extract}\right)\times 100$$

In order to obtain the encapsulation efficiency of the marjoram phenolic compounds in particles obtained by spray drying, 45 mg of the particles were dispersed in 5 mL of water and shaken for 10 min. Then, 500 µL of the supernatant (non-encapsulated compounds) was centrifuged at 16,000× *g* for 45 min using the Amicon filter 3kDa (VWR, USA) and analyzed by HPLC. The total and individual EE% were calculated according to Equations (2) and (3).

#### 2.2.5. Marjoram Phenolic Compounds Release from Micro/Nanoparticles

The study of the phenolic compounds’ release from chitosan particles was performed following the method described by Da Silva et al. [16], with slights modifications.

For this purpose, 45 mg of the freeze-dried particles obtained by ionic gelation or 45 mg of spray-dried particles were suspended in 5 mL of phosphate-buffered saline (PBS) solution at pH 7.4 and 2. The suspensions were placed into a water bath at 37 °C under agitation. At 1, 2, and 3 h, 350 µL of the solution was collected from each tube and centrifuged with the Amicon filter 3 kDa (VWR, USA) at 16,000× *g* for 45 min. Then, the supernatants were analyzed by HPLC and the release of phenolic compounds in each time was calculated.

#### 2.2.6. In Vitro Simulated Gastrointestinal Digestion 

The oral, gastric, and intestinal phases of the digestion were carried out employing a titrator Titrino Plus 877 (Methrom AG, Herisau, Switzerland), following the method described by Villalva et al. [10]. Briefly, for the oral phase, 100 mg of particles were mixed with 5 mL of deionized water and 100 µL of α-amylase from human saliva type XIIIA (Sigma-Aldrich, St. Louis, MO, USA), then shaken for 2 min at 37 °C. Then, 25 mL of acid water (pH 2) and 127 mg of porcine pepsin from porcine mucosa (536 U/mg) (Sigma-Aldrich, St. Louis, MO, USA) were added and pH was adjusted to 2 with 0.1 M HCl and shaking for 1 h at 37 °C in darkness. After gastric digestion, intestinal digestion started by adjusting the pH of the digestion solution at 7.5 with 1 M NaOH. Then, 0.463 mL of 325 mM CaCl_2_ (Sigma-Aldrich, Madrid, Spain) and 1.389 mL of 3.25 M NaCl (Sigma-Aldrich, Madrid, Spain) were added. Subsequently, a pancreatic-bile extract consisting of 115.7 mg of bile salts (Sigma-Aldrich, St. Louis, MO, USA) in 2.8 mL of 3.32 mg/mL of pancreatin (Sigma-Aldrich, St. Louis, MO, USA) dissolved in 10 mM trizme-maleate buffer was added and incubated for 2 h at 37 °C in darkness, controlling the pH value at 7.5. After the 2 h, the sample was cooled to inactivate the enzymes and centrifuged at 11,000 rpm for 15 min; the supernatant obtained was freeze-dried, then dissolved in DMSO and centrifuged for 40 min at 16,000× *g* by the Amicon filter 3 kDa (VWR, USA) to eliminate any remaining chitosan; then, it was filtered with a 0.45 µm PVDF filter and analyzed by HPLC-PAD.

### 2.3. Statistical Analysis

The statistical analysis was carried out using Statgraphics v. Centurion XVI software for Windows (Statpoint Inc., Warranton, VA, USA). The one-way analysis of variance (ANOVA) was used following Fisher’s least significant difference test (LSD) with a *p* ≤ 0.05.

## 3. Results and Discussion

### 3.1. Phenolic Characterization of UAE Marjoram Extract

UAE marjoram extract was obtained using ethanol in a ratio of 1:10 (plant: solvent) and submitted to ultrasound for 20 min. These extraction conditions were supported by previous studies in order to obtain an extract with a high content of phenolic compounds. The phenolic-compounds characterization of this extract was performed by HPLC-PAD analysis (Figure 1). As shown in Table 1, 29 compounds were identified, with arbutin and rosmarinic acid (RA) being the most abundant compounds, with lithospermic acid isomer and vicenin II in a lesser amount In addition, the quantities of luteolin, apigenin, and apigenin derivatives presented in the extract were remarkable. These results were consistent with those previously reported by other authors for *Origanum majorana* L. UAE extracts [17,18].

### 3.2. Formulation and Characterization of Micro/Nanoparticles Obtained by Ionic Gelation

#### 3.2.1. Formulation of Particles 

The formulation process of chitosan particles by ionic-gelation-adding TPP was influenced by both the chitosan and the TPP concentrations. Therefore, preliminary experiments were carried out to optimize the chitosan:TPP ratio for the formulation of particles. The chitosan:TPP ratios assayed in this work were 5:1, 6:1, and 7:1. Results showed that the ratio 7:1 did not produce particles with when both LCH chitosan and MCH chitosan were used. This result could be explained because the amount of chitosan employed was too high in comparison with TPP, as the formulation of micro/nanoparticles by ionic gelation is produced by the intermolecular crosslinking of chitosan amino groups that are positively charged and the TPP negatively charged phosphate group [19]. 

In addition, different marjoram extract concentrations (3, 2, 1, 0.5, and 0.25 mg/mL) were tested to optimize the formulation of particles. The results showed that when employing the highest extract concentrations (3 and 2 mg/mL), particles were not produced. This fact could be related to the addition of an excessive amount of extract in relation to the quantity of chitosan:TPP.

Therefore, the two chitosan:TPP ratios (5:1 and 6:1) and three marjoram extract concentrations (1, 0.5, and 0.25 mg/mL) for both LCH and MCH were selected for further assays. The formulation codes are shown in Table 2.

#### 3.2.2. Yields

Regarding the yields of the particle-formulation process (Table 2), it can be observed that the values increased when using MCH and a molecular CH:TPP ratio of 6:1. In addition, when increasing the quantity of extract, the yield value increased. Thus, the formulation IG12 (MCH, CH:TPP ratio 6:1 and 1 mg/mL of extract) presented the highest-yield value (70%). This result could be explained, as this formulation contained the largest quantity of solids [17]. Thus, yield values obtained were near to those reported by Cerchiara et al. [19] for chitosan nanoparticles prepared with similar chitosan:TTP ratios.

#### 3.2.3. Encapsulation Efficiency (EE)

The EE values for all of the formulations developed (Table 2) showed that this value was highly related to the concentration of the extract used. Thus, when using 1 mg/mL of extract, the EE values were much lower than they were with 0.25 and 0.5 mg/mL of extract. Formulations obtained with 0.25 and 0.5 mg/mL presented values from 54.4 to 61.0%. This result might be related to the competitive interaction between the phenolic –OH and phosphate groups of TPP for chitosan-protonated amino groups [20], as the formulations with high concentrations of phenolic compounds presented low EE values. Regarding the influence of the chitosan:TPP ratio, the EE values obtained showed no significative differences between using the 6:1 or the 5:1 chitosan:TPP ratios with 0.25 or 0.5 mg/mL of extract. Furthermore, no significant differences were found when chitosan of different molecular weight was used. 

Thus, the formulations prepared with 0.25 and 0.5 mg/mL of extract presented the highest EE values. However, formulations with 0.5 mg/mL allowed us to incorporate a higher extract quantity, so they were selected for further assays (IG2, IG5, IG8, and IG11).

#### 3.2.4. Particle Size, Zeta Potential, and Polydispersity Index (PDI)

The particle size of selected formulations (Table 2) ranged from 625.7 to 780.1 nm. These results showed no significative differences in particle sizes when using the 6:1 or the 5:1 chitosan:TPP ratios. However, particles obtained with LCH presented a lower particle size than particles obtained with MCH. Kunjachan et al. [21] indicated that when the molecular weight of chitosan is increased, the particle size generally increased. The PDI values of the samples were between 0.69 and 0.48 (Table 2), indicating a uniform distribution of particles [22].

Zeta potential values are also shown in Table 2. All formulations presented a positive zeta potential, ranged from 24.3 to 28.9 mV. These values indicated the presence of chitosan on the particle surface and ensured good physical stability of the dispersions, due to electrostatic repulsion. It is known that zeta potential values of around ±20–30 mV indicate that the created colloidal system is stable in time and that the chitosan amino groups are on the microparticle surface [23].

### 3.3. Formulation and Characterization of Microparticles Obtained by Spray Drying

#### 3.3.1. The Formulation of Microparticles

The viscosity of biopolymer suspension has been reported as an important parameter in order to obtain microparticles by the spray-drying technique. Suspensions with a high viscosity may block the atomizer and decrease the encapsulation yield [24]. Thus, preliminary assays using different LCH and MCH (2, 1, 0.75, and 0.5% *w*/*v*) and acetic acid (1, 0.5, and 0.4% *v*/*v*) concentrations were carried out to determine the optimal suspension viscosity. The results showed that 0.75% of LCH dissolved in 0.4% of acetic acid presented the best conditions for further studies. 

In addition, three different marjoram extract concentrations, expressed as LCH:extract ratios (4:1, 6:1, and 8:1), were also tested to optimize the formulation of particles. Formulation codes are shown in Table 3.

#### 3.3.2. Yields

The yields of the microparticles formulation process (Table 3) ranged from 24.2 to 31.5%. A trend in increasing yield was observed with the increase of the polymer in the polymer:extract ratio. These relatively low-yield values are frequent in chitosan microparticles prepared by the spray-drying method in laboratorial scale and they could be attributed to the small amount of material employed and the loss of the smallest particles during the process [25,26]. Thus, Cabral et al. [17] indicated that chitosan microparticles produced by spray drying presented yields equal to or less than 50%. However, Gelfuso et al. [27] indicated that the use of larger volumes of solids, as in industrial production, could improve the yields obtained.

#### 3.3.3. Encapsulation Efficiency (EE), Particle Size, Zeta Potential, and the Polydispersity Index (PDI)

EE values from 54.8 to 75.8% are shown in Table 3. The 4:1 and 6:1 LCH:extract ratios presented the highest EE%; meanwhile, when using the ratio 8:1, the EE value decreased. Similar results were reported by [17] when employing a 6:1 chitosan:Jabuticaba extract ratio, following the same encapsulation technique. In addition, these results showed that spray-dried microparticles presented higher EE% values with respect to microparticles obtained by ionic gelation. Considering the EE values, SP1 and SP2 formulations were selected for further assays.

Table 3 shows the particle size, zeta potential, and PDI of the particles that were formulated. Regarding particle size, no significative differences were found among the three formulations (the sizes ranged from 1.68 to 1.55 µm). In addition, the microparticles presented PDI values that indicated a good uniformity and respect to zeta potential and the values showed that the microparticles were stable.

### 3.4. EE of Marjoram Extract Phenolic Compounds

In order to evaluate the EE of phenolic compounds previously described as potential agents against colorectal cancer (mainly rosmarinic acid, luteolin, apigenin and apigenin’s derivatives), the measurement of the EE of individual phenolics presented in the extract was carried out.

Regarding microparticles obtained by ionic gelation, the results did not show significant differences between the uses of LCH or MCH, so only the MCH results are shown (Table 4). IG8 and IG11 formulations showed that the EE of the phenolic compounds studied was closely related to their polarity. Thus, for both formulations, the most polar compounds, such as arbutin, vicenin II, and luteolin 7-O-β-glucuronide, presented the lowest percentage of encapsulation (<50%). By decreasing phenolic compound polarity, its percentage of encapsulation increased, up to 100% for the most nonpolar compounds (salvianolic acid, eriodyctiol, luteolin, apigenin, apigenin derivative, and the last flavanone). Table 4 data also show that formulation IG8 (MCH:TPP ratio 5:1) presented a higher EE for arbutin, vicenin II, and luteolin 7-O-β-glucuronide than IG 11 formulation. However, rosmarinic acid showed a similar EE% in both formulations, of close to 55%. This value was closer to previous data reported for rosmarinic acid EE in chitosan particles obtained by ionic gelation [19]. Luteolin and apigenin and apigenin’s less polar derivatives presented 100% of the EE in both formulations.

With respect to spray-dried microparticles (Table 4), the results also indicated that the EE of the phenolic compounds was related to their polarity. However, the SP1 and SP2 formulations presented, for most compounds, higher EE percentages than formulations obtained by ionic gelation. Thus, arbutin and vicenin II showed values close to 60%. In addition, the EE percentage for rosmarinic acid was higher than 96% in both formulations.

Considering these results, microparticles obtained by spray drying (SP1 and SP2 formulations) were chosen for controlled release experiments.

### 3.5. Controlled Release Studies

The study of the release profile for the different phenolic compounds in the SP1 and SP2 formulations was performed in two different media, a phosphate-buffered saline (PBS) solution at pH 2 and another at pH 7.4, in order to simulate gastrointestinal pHs. 

The results at pH 2 (Table 5) were expressed as the amount (%) of the compound release at 1, 2, and 3 h. As shown in Table 5, at 1 h, an important initial burst release was obtained for the compounds between arbutin and rosmarinic acid, followed by a constant sustained release at 2 and 3 h. Arbutin presented the highest initial release (75%), following by vicenin II (54%); for the rest of the compounds, the values ranged from 2.5 to 36%. Compounds that were included between salvianolic acid and the flavanone remained in the microparticles at 3 h (release 0%). Because this initial release has been related to the desorption of compounds located on the surface or in the outer layers of the particle [26], it could be concluded that the more polar extract compounds, such as arbutin and vicenin II, seemed to be mainly located on or near the particle surface. As the polarity of phenolic compounds decreased, their initial release also decreased, indicating a longer diffusional distance between their location and the receptor solution. However, this higher release of polar compounds could be also influenced by the affinity of these compounds with the medium employed to carry out the release experiments.

The phenolics release profile at pH 7.4 (Table 5) from SP1 and SP2 formulations also exhibited a high initial burst (1 h), mainly for arbutin and vicenin II, followed by a sustained release at 2 and 3 h. As obtained at pH 2, reducing the polarity of the compounds decreased their initial release from the particles. However, at pH 7.4, the amount (%) of several compounds released at 3 h was lower than the amount at pH 2. These results were in concordance with previous data and could be related with the better solubility of chitosan in pH acid. Harris et al. [28] encapsulated polyphenols in chitosan microparticles by spray drying and reported 60% of polyphenols released from particles after 4 h at pH 5.7 and between 40–45% of polyphenols released at pH 6.5, indicating a better solubility of chitosan at lower pH.

Comparing the SP1 and SP2 formulations, both presented a similar release profile for most of the compounds in both pHs. However, at pH 2, the SP1 formulation presented a higher release for several compounds, such as rosmarinic acid derivative, eriodyctiol, apigenin derivative, and the last flavanone, in comparison with that of the SP2 formulation. This fact could be attributed to the higher quantity of extract in the SPI formulation. Considering these results, the SP2 formulation was chosen to carry out an in vitro digestion.

### 3.6. In Vitro Simulated Gastrointestinal Digestion

Finally, an in vitro gastrointestinal digestion was carried out to study the combined effect of pH and gastrointestinal enzymes on the developed SP2 formulation. 

The amount (%) of phenolics released after the gastrointestinal digestion process (Figure 2) indicated that arbutin presented the highest release (67%), followed by vicenin II (55%); for the rest of the compounds, the values ranged from 1.3% to 34.2%. These results were similar to those obtained in the release kinetics at different pHs for compounds between arbutin and rosmarinic acid. However, for compounds between lithospermic acid and the last flavanone, a release range from 6.2% to 24.5% was obtained, while in the previous kinetics, the release from the particles was very small, or 0%. In this sense, it must be taken into account that the digestion media include gastrointestinal enzymes. These enzymes could affect the phenolic compounds released and modify their structures [29].

## 4. Conclusions

In this study, chitosan micro/nanoparticles loaded with marjoram extract were prepared using two different techniques: ionic gelation and spray drying. The spray drying of microparticles showed the best performance in terms of encapsulation efficiency (75%) but were larger in size (1.55–1.68 µm). This technique also allowed the encapsulation of larger quantities of extract (1.25–1.88 mg/mL) and, after gastrointestinal digestion, most of marjoram phenolic compounds remained encapsulated in the chitosan spray-dried microparticles. 

Thus, chitosan microparticles prepared using spray drying as the encapsulation technique could be proposed as colon-specific systems for marjoram phenolic compounds.

## Figures and Tables

**Figure 1 foods-11-03657-f001:**
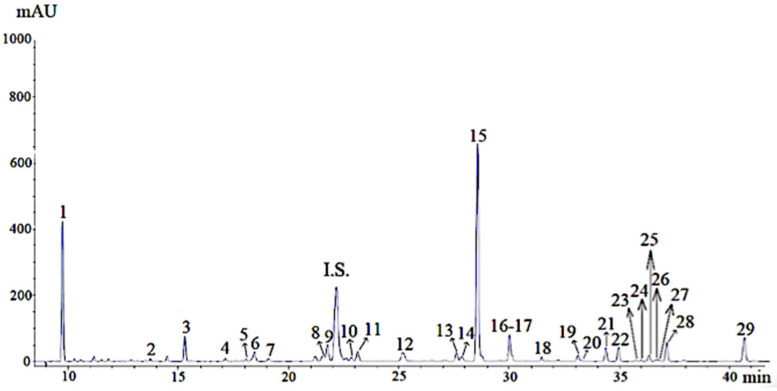
Chromatogram of phenolic compounds of UAE marjoram extract. Note: The numbers of the peaks correspond to the compounds listed in Table 1. I.S.: internal standard (Ethyl gallate).

**Figure 2 foods-11-03657-f002:**
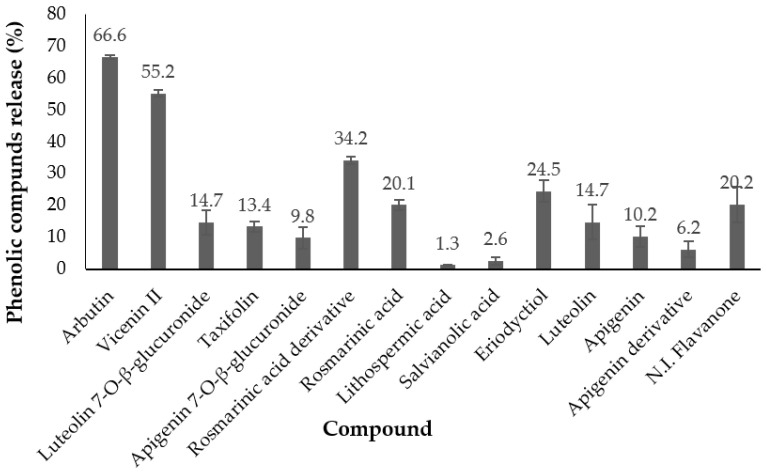
Amount (%) of phenolic compounds released after in vitro gastrointestinal digestion (mean ± SD, *n* = 3).

**Table 1 foods-11-03657-t001:** Phenolic composition of UAE marjoram extract.

Peak	Rt (min)	Compound	Concentration (mg/g)
1	9.8	Arbutin *	78.10 ± 1.18
2	13.7	Luteolin dihexoside	0.20 ± 0.01
3	15.3	Vicenin II *	4.56 ± 0.10
4	17.1	Caffeic acid *	0.47 ± 0.01
5	18.1	Luteolin monohexoside	0.42 ± 0.01
6	18.4	2,5 Hydroxybenzoic acid	3.23 ± 0.19
7	19.1	6-hydroxyluteolin-7-O-glucoside	0.76 ± 0.01
8	21.6	Apigenin derivative	0.82 ± 0.02
9	22.0	Caffeic acid derivative	1.91 ± 0.03
10	22.8	Luteolin 7-O-β-glucoside *	1.06 ± 0.08
11	23.1	Luteolin 7-O-β-glucuronide *	3.27 ± 0.09
12	25.2	Taxifolin *	2.33 ± 0.03
13	27.6	Apigenin 7-O-β-glucuronide *	2.24 ± 0.04
14	27.8	Rosmarinic acid derivative	1.99 ± 0.01
15	28.0	Rosmarinic acid *	48.32 ± 0.70
16	30.0	Lithospermic acid isomer	8.48 ± 0.10
17	30.2	Salvianolic acid isomer	2.09 ± 0.08
18	31.5	Rosmarinic acid derivative	1.69 ± 0.03
19	33.1	Eriodyctiol *	1.26 ± 0.01
20	33.3	Luteolin *	0.21 ± 0.00
21	34.4	N.I. Flavanone	1.78 ± 0.01
22	35.0	Apigenin methoxylated	1.34 ± 0.02
23	35.8	N. I. Flavanone	0.24 ± 0.00
24	36.0	Rosmarinic acid derivative	1.07 ± 0.04
25	36.3	N.I. Flavanone	1.02 ± 0.01
26	36.7	Apigenin *	0.21 ± 0.00
27	36.8	Naringenin *	0.34 ± 0.00
28	37.2	Apigenin derivative	1.29 ± 0.03
29	40.7	N.I. Flavanone	3.51 ± 0.05
		Σ Phenolic compounds	174.18 ± 2.89

* Comparison with standard. Rt: retention time. N.I.: non-identified.

**Table 2 foods-11-03657-t002:** Characterization of different formulations obtained by ionic gelation (mean ± SD, *n* = 3). LCH: chitosan low molecular weight; MCH: chitosan medium molecular weight.

Ratio CH:TPP	Formulation Code	Extract (mg/mL)	Yield (%)	EE (%)	Particle Size (nm)	ζ-Potential (mV)	PDI
**LCH 5:1**	IG1	0.25	32.0 ± 0.1 ^h^	59.3 ± 4.1 ^a^	-	-	-
**LCH 5:1**	IG2	0.5	36.8 ± 0.3 ^gh^	54.4 ± 0.6 ^a^	625.7 ± 54.7 ^b^	27.6 ± 0.7 ^a^	0.50 ± 0.02 ^bc^
**LCH 5:1**	IG3	1	39.1 ± 0.9 ^fgh^	25.3 ± 3.2 ^b^	-	-	-
**LCH 6:1**	IG4	0.25	42.1 ± 1.5 ^efg^	61.0 ± 7.2 ^a^	-	-	-
**LCH 6:1**	IG5	0.5	44.6 ± 6.2 ^def^	57.0 ± 2.8 ^a^	641.6 ± 53.6 ^b^	26.9 ± 0.9 ^a^	0.55 ± 0.04 ^b^
**LCH 6:1**	IG6	1	45.8 ± 2.9 ^de^	9.5 ± 0.8 ^d^	-	-	-
**MCH 5:1**	IG7	0.25	40.8 ± 2.6 ^efg^	58.3 ± 1.5 ^a^	-	-	-
**MCH 5:1**	IG8	0.5	49.2 ± 3.9 ^cd^	60.7 ± 5.5 ^a^	770.7 ± 30.1 ^a^	24.3 ± 1.7	0.69 ± 0.02 ^a^
**MCH 5:1**	IG9	1	55.1 ± 0.1 ^bc^	17.5 ± 2.6 ^c^	-	-	-
**MCH 6:1**	IG10	0.25	56.0 ± 1.4 ^bc^	60.8 ± 3.2 a	-	-	-
**MCH 6:1**	IG11	0.5	60.4 ± 3.7 ^b^	57.7 ± 1.0 ^a^	780.1 ± 24.1 ^a^	28.9 ± 1.6 ^a^	0.48 ± 0.03 ^c^
**MCH 6:1**	IG12	1	70.1 ± 5.8 ^a^	9.3 ± 1.0 ^d^	-	-	-

^a–h^ Different superscript letters denote significant differences within the same column.

**Table 3 foods-11-03657-t003:** Characterization of formulations obtained by spray drying (mean ± SD, *n* = 3). Relation CH:ext: relation chitosan:extract.

Code	CH (mg/mL)	Extract (mg/mL)	Relation CH:ext	Yield (%)	EE (%)	Particle Size (μm)	ζ-Potential (mV)	PDI
SP1	7.52	1.88	4:1	24.2 ± 1.7 ^b^	75.8 ± 5.3 ^a^	1.55 ± 0.16 ^a^	34.3 ± 0.7 ^a^	0.49 ± 0.04 ^a^
SP2	7.52	1.25	6:1	26.9 ± 2.5 ^b^	73.2 ± 6.1 ^a^	1.68 ± 0.44 ^a^	29.2 ± 0.6 ^b^	0.56 ± 0.07 ^a^
SP3	7.52	0.94	8:1	31.5 ± 0.7 ^a^	54.8 ± 0.9 ^b^	1.65 ± 0.11 ^a^	31.4 ± 0.6 ^b^	0.47 ± 0.03 ^a^

^a, b^ Different superscript letters denote significant differences within the same column.

**Table 4 foods-11-03657-t004:** Encapsulation efficiency (%) of different phenolic compounds in formulations IG8, IG11, SP1, and SP2 (mean ± SD, *n* = 3).

Compounds	Ionic Gelation		Spray Drying	
	IG8	IG11	SP1	SP2
Arbutin	45.26 ± 7.35 ^a^	34.94 ± 6.75 ^b^	63.06 ± 2.04	58.95 ± 7.33
Vicenin II	43.77 ± 6.14 ^a^	32.81 ± 5.66 ^b^	63.50 ± 7.72	62.61 ± 10.94
Luteolin 7-O-β-glucuronide	49.93 ± 7.45 ^a^	37.25 ± 3.42 ^b^	100 ± 0.00	100 ± 0.00
Taxifolin	76.41 ± 8.52	73.59 ± 5.79	91.05 ± 7.57	93.44 ± 4.86
Apigenin 7-O-β-glucuronide	58.76 ± 6.37	53.24 ± 6.37	100 ± 0.00	100 ± 0.00
Rosmarinic acid derivative	60.63 ± 5.97	54.44 ± 5.85	83.07 ± 1.51	79.39 ± 2.28
Rosmarinic acid	55.84 ± 7.95	50.11 ± 5.80	96.40 ± 1.84	96.88 ± 1.85
Lithospermic acid	87.53 ± 4.89	86.93 ± 4.68	100 ± 0.00	100 ± 0.00
Salvianolic acid	100 ± 0.00	100 ± 0.00	100 ± 0.00	100 ± 0.00
Eriodyctiol	100 ± 0.00	100 ± 0.00	100 ± 0.00	100 ± 0.00
Luteolin	100 ± 0.00	100 ± 0.00	100 ± 0.00	100 ± 0.00
Apigenin	100 ± 0.00	100 ± 0.00	100 ± 0.00	100 ± 0.00
Apigenin derivative	100 ± 0.00	100 ± 0.00	100 ± 0.00	100 ± 0.00
N. I. Flavanone	100 ± 0.00	100 ± 0.00	100 ± 0.00	100 ± 0.00

^a, b^ Different superscript letters denote significant differences within the same column.

**Table 5 foods-11-03657-t005:** Amount (%) of phenolic compounds released after 1, 2, and 3 h from SP1 and SP2 formulations at pH 2 and pH 7.4 (mean ± SD, *n* = 3).

Compounds	Amount (%) Release at pH 2					
	SP1			SP2		
	1 h	2 h	3 h	1 h	2 h	3 h
Arbutin	75.0 ± 5.6	84.1 ± 5.7	86.7 ± 9.9	75.4 ± 3.5	76.8 ± 4.5	84.4 ± 3.4
Vicenin II	53.9 ± 5.4	60.7 ± 7.6	70.2 ± 7.8	51.3 ± 2.8	59.9 ± 7.8	66.6 ± 2.2
Luteolin 7-O-β-glucuronide	35.1 ± 1.5 ^a^	46.2 ± 3.1	57.8 ± 3.6	30.5 ± 2.6 ^b^	46.3 ± 5.2	56.0 ± 2.4
Taxifolin	34.2 ± 6.4	34.8 ± 7.5	51.7 ± 9.9	25.4 ± 5.6	32.6 ± 9.9	41.1 ± 9.2
Apigenin 7-O-β-glucuronide	29.2 ± 0.3	31.6 ± 3.7	44.4 ± 10.8	25.8 ± 7.0	27.4 ± 3.9	32.7 ± 6.9
Rosmarinic acid derivative	36.0 ± 6.5	42.1 ± 7.3	55.4 ± 2.6 ^a^	28.5 ± 8.2	31.1 ± 5.8	34.4 ± 7.5 ^b^
Rosmarinic acid	32.7 ± 6.3	39.6 ± 4.7	54.8 ± 5.9	27.5 ± 9.6	38.7 ± 4.2	49.7 ± 5.6
Lithospermic acid	2.5 ± 1.3	2.8 ± 1.1	2.9 ± 1.2	2.9 ± 0.1	3.8 ± 1.4	5.1 ± 3.5
Salvianolic acid	0 ± 0	13.8 ± 1.7	14.8 ± 3.1	0 ± 0	12.1 ± 0.4	15.3 ± 5.1
Eriodyctiol	0 ± 0	3.5 ± 1.8 ^a^	6.8 ± 2.5 ^a^	0 ± 0	0 ± 0 ^b^	0 ± 0 ^b^
Luteolin	0 ± 0	0 ± 0	0 ± 0	0 ± 0	0 ± 0	0 ± 0
Apigenin	0 ± 0	0 ± 0	0 ± 0	0 ± 0	0 ± 0	0 ± 0
Apigenin derivative	0 ± 0	0 ± 0	4.2 ± 1.9 ^a^	0 ± 0	0 ± 0	0 ± 0
N. I. Flavanone	0 ± 0	2.8 ± 1.1 ^a^	5.1 ± 3.2 ^a^	0 ± 0	0.1 ± 0 ^b^	1.7 ± 0.1 ^b^
**Compounds**	**Amount (%) Release at pH 7.4**					
	**SP1**			**SP2**		
	**1 h**	**2 h**	**3 h**	**1 h**	**2 h**	**3 h**
Arbutin	64.9 ± 8.3	80.9 ± 7.8	84.9 ± 4.4	53.6 ± 9.7	77.1 ± 1.8	79.7 ± 7.7
Vicenin II	44.9 ± 6.1	60.9 ± 6.2	72.7 ± 1.6	35.8 ± 8.2	58.5 ± 3.6	63.4 ± 7.4
Luteolin 7-O-β-glucuronide	23.9 ± 1.2	34.7 ± 1.6 ^a^	36.3 ± 2.8	23.5 ± 1.3	29.5 ± 0.7 ^b^	34.9 ± 5.6
Taxifolin	26.4 ± 2.9	29.7 ± 7.9	32.7 ± 7.6	26.0 ± 3.9	29.9 ± 3.1	31.8 ± 8.7
Apigenin 7-O-β-glucuronide	19.2 ± 1.8	22.6 ± 4.1	29.1 ± 6.9	16.9 ± 3.3	20.5 ± 1.3	26.6 ± 6.9
Rosmarinic acid derivative	29.8 ± 1.6	30.5 ± 2.5	39.4 ± 9.1	27.6 ± 4.6	32.8 ± 2.5	33.8 ± 6.2
Rosmarinic acid	30.7 ± 8.1	46.8 ± 2.2	49.8 ± 9.5	28.4 ± 9.4	35.8 ± 8.3	41.6 ± 8.1
Lithospermic acid	2.5 ± 0.9 ^a^	2.8 ± 1.1	2.9 ± 1.2	0 ± 0 ^b^	1.3 ± 0.7	1.9 ± 0.6
Salvianolic acid	0 ± 0	0 ± 0	0 ± 0	0 ± 0	0 ± 0	0 ± 0
Eriodyctiol	0 ± 0	0 ± 0	3.0 ± 1.2	0 ± 0	0 ± 0	2.2 ± 1.4
Luteolin	0 ± 0	0 ± 0	0 ± 0	0 ± 0	0 ± 0	0 ± 0
Apigenin	0 ± 0	0 ± 0	0 ± 0	0 ± 0	0 ± 0	0 ± 0
Apigenin derivative	0 ± 0	0 ± 0	1.9 ± 0.6	0 ± 0	0 ± 0	1.4 ± 0.5
N.I. Flavanone	0 ± 0	1.4 ± 0.6	2.8 ± 1.1	0 ± 0	1.3 ± 0.5	2.5 ± 1.3

^a, b^ Different superscript letters denote significant differences between SP1 and SP2 formulations in the same pH and same release time.

## Data Availability

Data is contained within the article.

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
