# Peer review of "Encapsulation of Marjoram Phenolic Compounds Using Chitosan to Improve Its Colon Delivery"

_foods, 2022, doi:10.3390/foods11223657_

Round 1

Reviewer 1 Report

In this manuscript authors have discussed the results on the preparation of CH micro/nanoparticles and its application to increase the colon-specific delivery of marjoram phenolic compounds. Overall the manuscript is presented well and the flow of manuscript is interesting. However, there are some major issues which must be taken care of.

1.      How authors have confirmed the authenticity of plant material used in this study.

2.      Authors should confirm the authenticity of plant material from a plant taxonomist as it is difficult to identify the species of a plant directly purchased from the local shop.

3.      Why authors used ethanol as extracting solvent?

4.      Why the plant material was not defatted prior to its extraction with ethanol?

5.      Authors must include the HPLC chromatogram of the plant extract and standard compounds in the manuscript.

6.      How quantification of identified compounds was achieved?

7.      In Table 1 authors have identified derivatives of some specific compounds for example Caffeic acid derivative, Rosmarinic acid derivative etc. How were identification of these compounds confirmed.

8.      Authors should screen the bioactivity of the plant extract. Specially the anticancer properties of the extract used in this study.

Author Response

1.- How authors have confirmed the authenticity of plant material used in this study.

Plant material was supplied by Murciana de Herboristería S.A. Murciana de Herboristería S.A., a company located in Murcia (Spain) and dedicated since 1989 to the harvesting, conditioning and wholesale marketing of condiments and spices, plant species for infusion, teas and medicinal plants for food and pharmaceutical use.

When you buy plant material to this company, they always provide you a certificate of analysis, including the authenticity of plant material, information about the cultivation and collection, organoleptic and physical-chemical characteristics. In this case, the certificate indicated that the plant material provided was Origanum majorana.

2.- Authors should confirm the authenticity of plant material from a plant taxonomist as it is difficult to identify the species of a plant directly purchased from the local shop.

As I explained in the previous point, Murciana de Herboristería S.A., is a company specialising in the supply of plant material. Therefore, this company always provides you a certificate of plant material analysis, including the authenticity of this material.

To avoid confusion, we will replace in the manuscript “local herbalist shop” by “Spanish Company specialized in plant material supplies”.

3.- Why authors used ethanol as extracting solvent?

In previous studies carried out in this research group, the influence of the extraction solvent was studied. Several solvents such as acetone, hexane, ethyl acetate, methanol and ethanol were studied. The results showed that ethanol allowed us to extract an important quantity of phenolic compounds, using a solvent classified as GRAS (Generally Recognized as Safe). The GRAS solvents present the advantage that they can be use in the development foodstuffs.

4.- Why the plant material was not defatted prior to its extraction with ethanol?

The plant material was not defatted prior to its extraction because we did not want to risk losing less polar phenolic compounds.

5.-Authors must include the HPLC chromatogram of the plant extract and standard compounds in the manuscript.

The HPLC chromatogram of the plant extract will be included in the manuscript as Figure 1 and standard compounds used for identification purpose has been marked with an asterisk in Table 1.

6.-How quantification of identified compounds was achieved?

Quantification of identified compounds was carried out by using calibration curves of its authentic standard at five levels in triplicate by HPLC−PAD. Compounds for which no standards were available were quantified with the calibration curve of the compound with the highest structural affinity, e.g., luteolin-7-O-glucoside was used for luteolin glycosylated derivative, homoorientin for luteolin dihexoside and luteolin monohexoside, caffeic acid for caffeic acid derivative and apigenin for apigenin methoxylated. All this information is specified in Villalva et al [10].

7.- In Table 1 authors have identified derivatives of some specific compounds for example Caffeic acid derivative, Rosmarinic acid derivative etc. How were identification of these compounds confirmed.

All the compounds were identified as previously reported by Villalva et al. (2021).

Villalva, M.; Santoyo, S.; Salas-Pérez, L.; Siles-Sánchez, M.N.; Rodríguez García-Risco, M.; Fornari, T.; Reglero, G.; Jaime, L. Sustainable extraction techniques for obtaining antioxidant and anti-Inflammatory compounds from the Lamiaceae and Asteraceae Species. Foods 2021, 10, 2067. https://doi.org/10.3390/foods10092067.

Briefly, the identification was carried out by a mass spectrometric detection using an HPLC system (1100 series, Agilent Technologies, Santa Clara, CA, USA) connected to an ultra-high-resolution QTOF instrument. In this regard, the identification of the phenolic compounds was performed by contrasting the accurate mass and MS/MS fragmentation pattern with the literature, and by comparison of its retention time and UV-Vis max absorption pattern with the available analytical standards.

8.- Authors should screen the bioactivity of the plant extract. Specially the anticancer properties of the extract used in this study.

Although the screening of the anticancer activity of this extract seems quite interesting, the aim of this study was only to develop Chitosan micro/nanoparticles to increase colon specific delivery of marjoram phenolic compounds. We indicated the potential use of this extract in colorectal cancer based in the literature. As indicated in the introduction section, rosmarinic acid, apigenin and lutein, compounds quantified in the marjoram extract have been reported to present anticancer activity. Thus, Venkatachalam et al. [12] reported that rosmarinic acid could act as a potent chemopreventive agent in colon cancer since it prevented the formation and multiplicity of aberrant crypt foci, protected cells from the damage caused by 1,2-dimethylhydrazine and its metabolites, spared the activities of antioxidant enzymes and induced apoptosis of tumor cells. Besides, Redondo-Blanco et al. [13] indicated that apigenin induced growth inhibition, cell cycle arrest and apoptosis in colon cancer lines cells. Luteolin has been also proposed as a therapeutic agent in colorectal tumors [14].

Reviewer 2 Report

The manuscript relies on the delivery of phenolic compounds encapsulated in chitosan. the study is interesting. Just I have two comments:

1) Why the authors decided extrac the phenolics by ultrasoun and not other method?

2) The section 2.2.6 describe the method to determination of gastrointestinal conditions. The authors comment that they following the method described by Villalva et al., 2018 with some modifications. Please provide the details about modifications, because it is important, as minor modifications can be impact in the results obtained. 

In my opinion the manuscript should be accepted after resolve "minor corrections" 

Author Response

1.-Why the authors decided to extract the phenolics by ultrasound and no other method?

Previous studies carried out in this research group using different extraction techniques (conventional solid-liquid extraction, pressurized liquid extraction (PLE) and ultrasound assisted extraction (UAE)) indicated that UAE allowed a higher recovery of phenolic compounds that conventional solid-liquid extraction and similar to PLE. Besides, UAE is considering a more efficient technique due to its solvent reduction consumption and shorter extraction times.

2.- The section 2.2.6 describe the method to determination of gastrointestinal conditions. The authors comment that they following the method described by Villalva et al., 2018 with some modifications. Please provide the details about modifications, because it is important, as minor modifications can be impact in the results obtained. 

The gastrointestinal conditions used in this work were identical to the method described by Villalva et al., 2018. The only modification was the preparation of the sample before the digestion process. In this case, 100 mg of microparticles were dissolved in 5 ml of water and in the previous paper 100 mg of the extract were dissolved in 5 ml of a mixture of 50% ethanol and 50% water.

We have included this modification in the manuscript.

Reviewer 3 Report

The manuscript entitled: "Chitosan based micro/nanoparticles for improving colon delivery of marjoram phenolic compounds" is about using chitosan as encapsulating material to improve colon delivery of marjoram phenolic compounds. In general, the manuscript is interesting and well-designed and the objectives of the research well fit the journal's aims and scopes. The methodology and research design well align with the research objective and the presentation style are clear. There are some comments to address by the authors before the final decision by the editor as follows:

1- Title: Revise the title, and make it more informative. A research title should clearly show the "Why" and "How" of the research. You need to bring "Encapsulation" in the title to make it more interesting.

2- Keywords: Better to choose keywords other than the main words of the title. It improves the visibility of the research.

3- Introduction: Background literature need to update and implement the recent publications in the field of Chitosan application; for instance this review article may improve your background: "Oladzadabbasabadi, Nazila, et al. "Recent advances in extraction, modification, and application of chitosan in packaging industry." Carbohydrate polymers 277 (2022): 118876." You can find a few related articles from Foods or other MDPI journals to update your literature.

4- About using CH as an abbreviation for Chitosan, I do not agree and recommend using the full name of chitosan.

5- Methodology: Please cite a proper reference for each method either a published article or a standard method. If you have developed the method, you need to bring all the details to show its validity.

6- Results and discussion, This part is almost OK, improve the discussion part on the control released and yield part, and add statistical analysis to Table 5.

Report some data in the figure to make the article more interesting for the readers.

7- Conclusion: Better to make it shorter, and focus on hypothesis justification and future research recommendations.

Author Response

1- Title: Revise the title, and make it more informative. A research title should clearly show the "Why" and "How" of the research. You need to bring "Encapsulation" in the title to make it more interesting.

As reviewer suggested, the title has been changed. The new title is “Encapsulation of marjoram phenolic compounds using chitosan for improving its colon delivery”

2- Keywords: Better to choose keywords other than the main words of the title. It improves the visibility of the research.

These new keywords have been proposed: Origanum majorana; phenolic compounds; nano/microparticles; colon delivery systems; ionic gelation; spray-drying

3- Introduction: Background literature need to update and implement the recent publications in the field of Chitosan application; for instance this review article may improve your background: "Oladzadabbasabadi, Nazila, et al. "Recent advances in extraction, modification, and application of chitosan in packaging industry." Carbohydrate polymers 277 (2022): 118876." You can find a few related articles from Foods or other MDPI journals to update your literature.

As reviewer suggested, background literature has been updating and recent publications in the field have been included.

  1. Oladzadabbasabadi, Nazila, et al. "Recent advances in extraction, modification, and application of chitosan in packaging industry."Carbohydrate polymers 277 (2022): 118876.
  2. Samprasit, W.; Opanasopit, P.; Chamsai, B. Alpha-mangostin and resveratrol, dual-drugs-loaded mucoadhesive thiolated chitosan-based nanoparticles for synergistic activity against colon cancer cells. Journal of biomedical material research, Part B (2022), 110, 1221-1233. https://doi.org/10.1002/jbm.b.34992.

4- About using CH as an abbreviation for Chitosan, I do not agree and recommend using the full name of chitosan.

This suggestion has been considered and CH has been substituted by Chitosan along all the manuscript.

5- Methodology: Please cite a proper reference for each method either a published article or a standard method. If you have developed the method, you need to bring all the details to show its validity.

We included in the manuscript a reference for extraction method:  the Ultrasound assisted extraction (UAE) was carried out following the method described in Villalva et al. (2021). Villalva, M.; Santoyo, S.; Salas-Pérez, L.; Siles-Sánchez, M.N.; Rodríguez García-Risco, M.; Fornari, T.; Reglero, G.; Jaime, L. Sustainable extraction techniques for obtaining antioxidant and anti-Inflammatory compounds from the Lamiaceae and Asteraceae Species. Foods 2021, 10, 2067.

We also included in the manuscript a reference for the study of the phenolic compounds releases from the particles. It was performed following the method described by Da Silva et al. (2013). Da Silva, S.B.; Oliveira, A.; Ferreira, D.; Sarmento, B.; Pintado, M. Development and validation method for simultaneous quantification of phenolic compounds in natural extracts and nanosystems. Phytochemical Analysis 2013, 24, 638–644. https://doi.org/10.1002/pca.2446

6- Results and discussion, this part is almost OK, improve the discussion part on the control released and yield part, and add statistical analysis to Table 5. Report some data in the figure to make the article more interesting for the readers.

Statistical analysis to Table 5 has been added and data showed in Table 6 has been included in a Figure (Figure 2).

The discussion on the yield part has been improved.  Text included in line 286: Thus, we obtained yield values close to those reported by Cerchiara et al. [19] for chitosan nanoparticles prepared with similar Chitosan:TTP ratios. Text included in line 347: Thus, Cabral et al. [17] indicated that Chitosan microparticles produced by spray drying presented yields equal or less than 50%.

The discussion on the control release part has been improved. Text included in line 425: Comparing the SP1 and SP2 formulations, both presented a similar release profile for most of the compounds in both pHs. However, at pH 2, SP1 formulation presented a higher release for several compounds like rosmarinic acid derivative, eriodyctiol, apigenin derivative and the last flavanone in comparison to SP2. This fact could be attributed to the higher quantity of extract in formulation SP1. Considering these results, SP2 formulation was chosen to carry out an in-vitro digestion.

7- Conclusion: Better to make it shorter, and focus on hypothesis justification and future research recommendations.

As reviewer suggested, conclusion has been made shorter and focus on hypothesis justification.

New conclusion: In this study, chitosan micro/nanoparticles loaded with marjoram extract were prepared using two different techniques: ionic gelation and spray-drying. Spray-drying microparticles showed the best performance in terms of encapsulation efficiency (75%) but presented the larger size (1.55-1.68µm). This technique also allowed to encapsulate larger quantities of extract (1.25-1.88 mg/mL) and after gastrointestinal digestion, most of marjoram phenolic compounds remained encapsulated in the microparticles.

Thus, chitosan microparticles prepared using spray-drying as encapsulation technique, could be proposed as colon-specific delivery systems for marjoram phenolic compounds.